# Temperate Agroforestry Systems and Insect Pollinators: A Review

**Gary Bentrup** [1],*, **Jennifer Hopwood** [2], **Nancy Lee Adamson** [3] and **Mace Vaughan** [4]

[1]  Department of Agriculture, U.S. Forest Service, National Agroforestry Center, Lincoln, NE 68583, USA
[2]  Xerces Society for Invertebrate Conservation, Omaha, NE 68134, USA; jennifer.hopwood@xerces.org
[3]  Xerces Society for Invertebrate Conservation, Greensboro, NC 27401, USA; nancy.adamson@xerces.org
[4]  Xerces Society for Invertebrate Conservation, Portland, OR 97232-1324, USA; mace.vaughan@xerces.org
*  Correspondence: gary.bentrup@usda.gov; Tel.: +1-(402)-437-5178

**Abstract:** Agroforestry can provide ecosystem services and benefits such as soil erosion control, microclimate modification for yield enhancement, economic diversification, livestock production and well-being, and water quality protection. Through increased structural and functional diversity in agricultural landscapes, agroforestry practices can also affect ecosystem services provided by insect pollinators. A literature review was conducted to synthesize information on how temperate agroforestry systems influence insect pollinators and their pollination services with particular focus on the role of trees and shrubs. Our review indicates that agroforestry practices can provide three overarching benefits for pollinators: (1) providing habitat including foraging resources and nesting or egg-laying sites, (2) enhancing site and landscape connectivity, and (3) mitigating pesticide exposure. In some cases, agroforestry practices may contribute to unintended consequences such as becoming a sink for pollinators, where they may have increased exposure to pesticide residue that can accumulate in agroforestry practices. Although there is some scientific evidence suggesting that agroforestry practices can enhance crop pollination and yield, more research needs to be conducted on a variety of crops to verify this ecosystem service. Through a more comprehensive understanding of the effects of agroforestry practices on pollinators and their key services, we can better design agroforestry systems to provide these benefits in addition to other desired ecosystem services.

**Keywords:** alley cropping; bees; forest farming; hedgerows; pollinators; pollination; riparian buffers; shelterbelts; windbreaks

## 1. Introduction

Plant pollination by animals is one of the most important ecosystem services and is essential in both natural and agricultural landscapes. An estimated 85% of the world's flowering plants depend on animals—mostly insects—for pollination [1]. Insect pollination is critical to food security and roughly 35% of global crop production is dependent on pollination by animals [2,3]. Pollinators are also a keystone group in most terrestrial ecosystems, necessary for plant reproduction, and important for wildlife food webs [4,5]. Insect pollinators include bees, wasps, flies, beetles, butterflies, and moths, but some bird and bat species pollinate as well [6]. Although all pollinators play important roles, bees are considered particularly essential for pollination of agricultural crops [7,8] as well as for wild plants in temperate climates [6]. Globally, insect pollinators are in decline, with some estimates that 40% of invertebrate pollinator species may be at risk of extinction worldwide [9]. Threats such as the loss, degradation, and fragmentation of habitat (e.g., [5,10,11]); introduced species (e.g., [12,13]); the use of pesticides (e.g., [4,14–16]; and diseases and parasites (e.g., [17,18]) all contribute to pollinator decline. An annual value of global crop output (estimated at $235 to $577 billion $US in 2015) is at

risk due to pollinator loss [9]. Threats to pollinators may have profound consequences for ecosystem health as well as our food systems [9,19,20].

Managing existing habitat and restoring additional habitat for pollinators has been demonstrated to increase their abundance and diversity (e.g., [13,21,22]). Agroforestry is the intentional integration of trees and/or shrubs with herbaceous crops and/or livestock in an agricultural production system [23]. By adding structural and functional diversity to agricultural landscapes, agroforestry can provide pollinator habitat and support pollinator services [24]. In temperate regions, agroforestry systems include many different permutations such as windbreaks, riparian buffers, alley cropping, hedgerows, shelterbelts, silvopasture, and forest farming [23]. Depending on the situation and application, these practices can provide protection for topsoil, livestock, and crops; increase crop and livestock productivity; reduce inputs of energy and chemicals; increase water use efficiency; improve air and water quality; sequester carbon; and enhance biodiversity [23,24]. Although agroforestry is rarely implemented for pollinator habitat or crop pollination services, there are opportunities to incorporate these services when designing multifunctional practices. A systematic review of existing scientific literature on the topic is a key first step. While several reviews have focused on biodiversity-based ecosystem services of agroforestry systems (e.g., [24,25]), none have examined in detail the effects of agroforestry on insect pollinators and pollination services. The objective of this paper is to review the role of temperate agroforestry systems in supporting insect pollinators and pollination services with a focus on the role of trees and shrubs in these systems.

## 2. Literature Review

Our systematic review followed recommended procedures outlined in the PRISMA statement [26]. An initial screening of research papers published in peer-reviewed English-language scientific journals was performed within Web of Science and Scopus databases. The literature was searched using a Boolean defined by logical strings containing one of the keywords "agroforestry", "alley cropping", "windbreak", "shelterbelt", "hedgerow", "forest farming", "multi-story cropping", "silvopasture", and "riparian buffer", in conjunction with at least one pollinator keyword. Pollinator keywords used in the queries included "bees", "flies", "moths", "wasps", "beetles", "pollinator", and "pollination". In order to have a comprehensive foundation for this review, the search was conducted for the earliest electronic records available to mid-2019. Review articles were included if they contained original research. We also examined reference lists within review articles to identify relevant papers.

To supplement this review, key articles on reducing pesticide drift and runoff using agroforestry practices were included even though some of these papers did not explicitly use pollinator terms (e.g., bees, butterflies, flies) in the text. It is recognized that reducing pesticide exposure pathways will have benefits for insect pollinators and we wanted to capture pesticide mitigation research regarding agroforestry practices. We also included papers that provided information on tree and shrub species used in temperate agroforestry practices where available. These papers were identified based on the authors' experience and knowledge regarding the topic area.

These initial records were screened for relevance to insect pollinators and pollination within temperate regions. These remaining articles were reviewed in whole and 134 were included in the final database. The PRISMA flowchart summarizing the search results and screening workflow is shown in Figure 1. Limiting our search to English articles in peer-reviewed sources in electronic databases exposes this review to the risk of language and publication bias [27]. To counter this potential bias, we included a broad variety of articles as possible to capture the dominant themes within the boundaries of the review objectives.

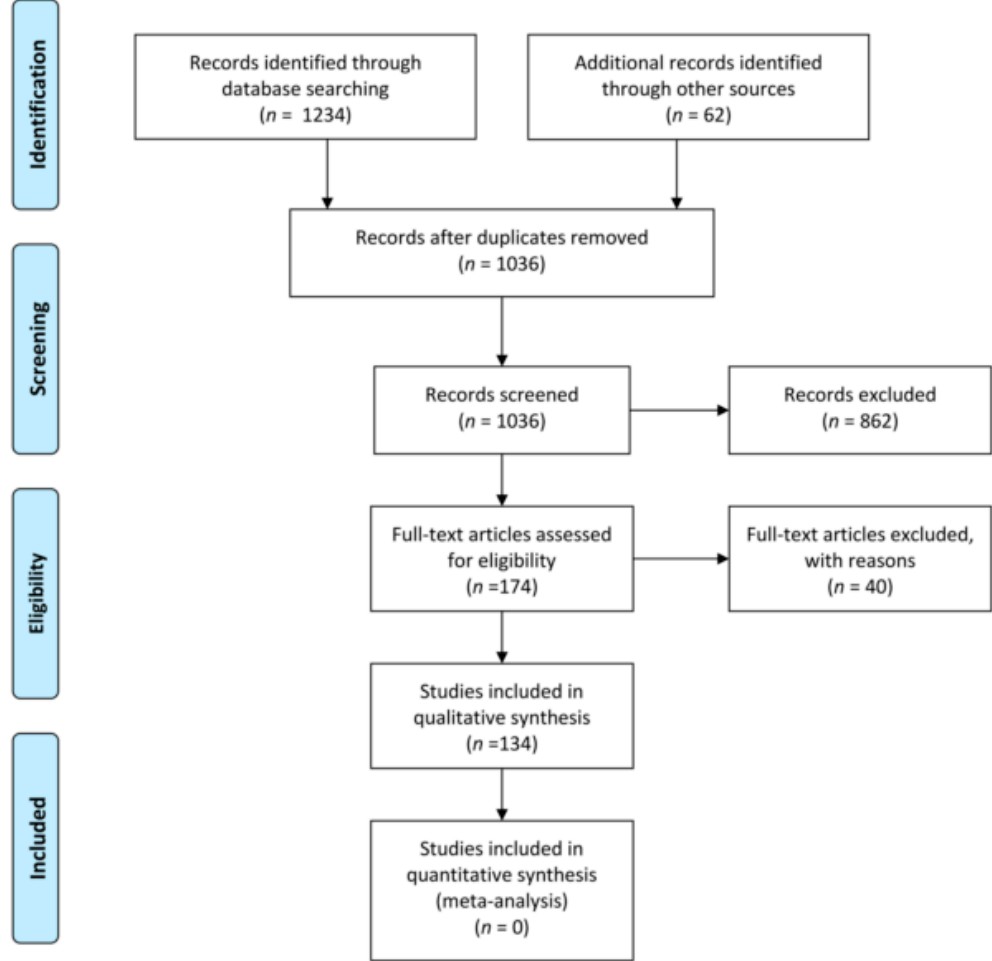

**Figure 1.** Flow diagram depicting the search and selection of the study process according to the Preferred Reporting Items for Systematic Reviews and Meta-Analyses (PRISMA) statement.

The majority of research articles in this review database was focused on windbreaks, shelterbelts, and hedgerows, with limited papers available on riparian buffers and alley cropping. There were few or no studies on silvopasture, forest farming, or multi-story cropping regarding insect pollinators in temperate regions. Our review indicates that agroforestry practices in general can provide three overarching functions for pollinators: (1) providing habitats, including foraging resources and nesting or egg-laying sites, (2) enhancing site and landscape connectivity, and (3) mitigating pesticide exposure. Available scientific evidence suggests that agroforestry practices can enhance crop pollination and yields; however, few studies have been conducted on this ecosystem service. Table 1 and Figure 2 provides a summary of this review.

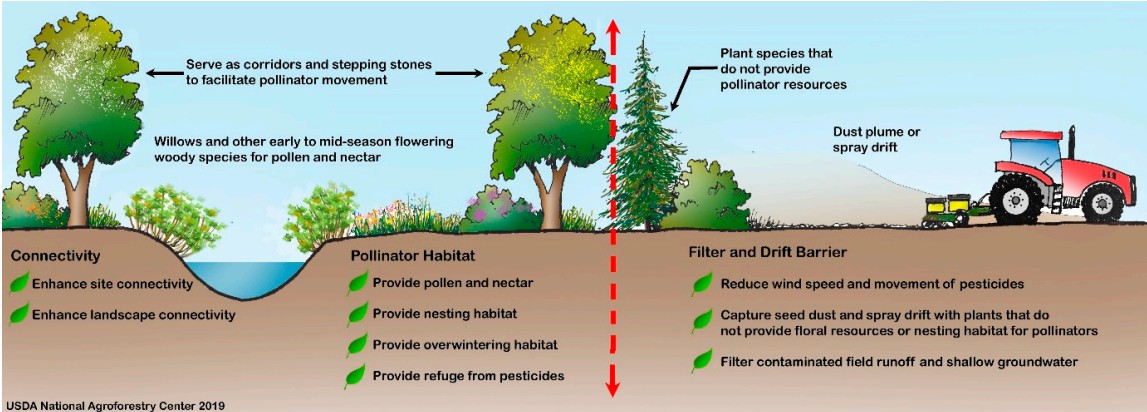

**Figure 2.** Conceptual diagram illustrating typical functions that a generic agroforestry practice can provide to insect pollinators.

## 3. Foraging Resources

### 3.1. Pollen and Nectar

Pollinators require a diversity of flowering plants over the foraging season to provide nectar and pollen resources to meet their nutritional needs [9]. Studies have documented diverse pollinator use of these floral resources in agroforestry practices, particularly if planted specifically to promote pollinators. Hedgerows, riparian buffers, windbreaks, and other linear agroforestry practices are used by bees [28–42], butterflies [36,37,42–49], moths [49–52], and flies [22,37,41,42,53,54]. Based on available evidence, woody species used in temperate agroforestry practices can play an important role in providing floral resources; however, if the agroforestry practice lacks pollinator-suitable floral resources, pollinator use is limited. For instance, Macdonald et al. [55] found limited pollinator use of shelterbelts in New Zealand that were predominantly comprised of Monterey pine (*Pinus radiata* D. Don) and Monterey cypress (*Hesperocyparis macrocarpa* (Hartw.) Bartel) (wind-pollinated exotic species).

Tree and shrub species offer abundant nectar with relatively high sugar contents such as maple (*Acer* spp.), horsechestnut (*Aesculus* spp.), basswood (*Tilia* spp.), willow (*Salix* spp.), brambles (*Rubus* spp.), cherry and plum (*Prunus* spp.), and serviceberry (*Amelanchier* spp.) [56–61]. For example, sugar content in horsechestnut (*Aesculus hippocastanum* L.) ranges from 0.58–3.57 mg/flower/24h, while black locust (*Robinia pseudoacacia* L.) ranges from 0.76–4.0 mg/flower/24h [62]. For comparison, white clover (*Trifolium repens* L.) ranges from 0.01–0.20 mg/flower/24h and alfalfa (*Medicago sativa* L.) ranges from 0.07–0.25 mg/flower/24h [62].

Pollen is a protein-rich resource that is used by native bees, honey bees, and some wasps to feed their brood or to provision their eggs. Pollen is also used by some adult and larval flies and beetles as a food source. Key woody species that can provide pollen with high concentrations of amino acids, sterols, trace minerals, and other nutritionally important compounds for bees and other pollinators include willow, maple, cherry and plum, brambles, chestnut (*Castanea* spp.), and ash (*Fraxinus* spp.) [60,61,63–66]. Some bees are pollen specialists (oligolectic), wholly dependent on specific shrubs and trees in certain families, such as willows, dogwoods (Cornaceae), heaths like blueberry and huckleberry (Ericaceae), buckthorns such as New Jersey tea (Rhamnaceae), and roses (Rosaceae) [67,68]. Other research has found apple (*Malus* spp.), hawthorn (*Crataegus* spp.), and elderberry (*Sambucus* spp.) to be attractive for pollinator foraging [31,33]. This list is not inclusive and additional woody species are likely to provide suitable forage resources even if they are not specifically mentioned in the literature reviewed.

Trees and shrubs in temperate regions often flower early in the spring and can deliver some of the first pollen and nectar resources of the season, boosting early-season pollinator populations [57,59–61,65,69,70]. In Michigan, US, Wood et al. [70] determined that willows, maples,

and *Prunus* spp. provided over 90% of the pollen collected in April by social and solitary bees. Flower density and subsequent nectar availability can be higher in some tree and shrub species compared to herbaceous species [57,58]. For instance, during peak flowering season, gray willow (*Salix cinerea* L.) can produce 334,178 flowers/m$^2$ and oneseed hawthorn (*Crataegus monogyna* Jacq.) 19,003 flowers/m$^2$ compared to sea aster (*Aster tripolium* L.) 9,565 flowers/m$^2$ and buttercup (*Ranunculus acris* L.) 688 flowers/m$^2$ [58]. Respectively, nectar productivity for these species is 3612, 584, 169, and 50 kg/ha cover/year. Spatially, agroforestry practices that include a diversity of flowering woody and herbaceous species can deliver a high density of floral resources relative to the land area occupied due to vertical layering [22,31,59,71,72]. One study documented approximately two and four times greater nectar per unit area in hedgerows compared to woodlands and pasture, respectively [35]. In regards to native versus exotic plants, one study showed that wild bees, and managed bees in some cases, prefer to forage on native plants in hedgerows over co-occurring weedy, exotic plants [28]. Management that reduces floral resources such as hedgerow trimming has been shown to have a negative impact on pollinators [73,74].

### 3.2. Resins and Oils

Bees also collect resins and floral oils from trees and other plants to aid in nest construction and provisioning immatures [75–78]. Some tunnel-nesting native bees use tree resins to seal off their nests [78], while other bees use tree resins and embedded sand grains and plant materials or debris to create nest cells [79]. Honey bees use resins to make propolis to seal unwanted holes in their hives [80]. Propolis has antibacterial properties that help reduce disease transmission and parasite invasion [81]. Poplar trees (*Populus* spp.) are a common source for these resins [82–85]. Due to their fast growth rate, low disease and pest issues, lower shading effect, and marketable products (i.e., biofeedstock, lumber), poplar trees are often used in temperate agroforestry applications. Other tree species that provide resin sources include: pine (*Pinus* spp.), birch (*Betula* spp.), elm (*Ulmus* spp.), alder (*Alnus* spp.), beech (*Fagus* spp.), and horsechestnut [82,84–86]. Depending on application, these tree species have traits or products that can be useful in agroforestry, such as being evergreen (pine), nitrogen-fixing (alder), syrup production (birch), lumber (elm, pine), mast (beech), and dense shade (horsechestnut).

### 3.3. Microclimate Modification

Pollinator behavior, foraging, and resulting pollination services are strongly influenced by weather conditions (e.g., ambient temperature, wind speed, precipitation) [87,88]. Temperature and wind speed are two primary weather variables that agroforestry practices can influence. Agroforestry practices can reduce air movement and modify temperatures in a cropped area. Daytime air temperatures are several degrees warmer within a certain distance downwind of windbreaks (8 to 10 times the windbreak height) [89]. These elevated temperatures can increase pollinator activity and pollination, particularly in vegetable- and fruit-growing regions, where air temperatures at pollination time can be below optimum [90,91]. The vertical structure and shaded sites found within tree-based practices may offer a diversity of niches that allow pollinators to find suitable sites for thermal regulation, which is becoming increasingly important under climate change [9]. One study found that agricultural landscapes that have a higher proportion of hedgerows and other semi-natural habitats (i.e., 17% compared to 2%) reduced the detrimental effects of warmer temperatures on native bee species' richness and abundance [92].

Additional thermoregulation considerations for managed honey bees may be addressed by agroforestry plantings. Honey bees expend energy to cool themselves and their hives during hot weather. If the hives are shaded, that energy can be diverted to honey production and hive maintenance activities. Trees and shrubs can be used to shade beehives, especially if the hives are placed on the north or northeast sides of the woody plantings to receive maximum shading during the summer heat [93]. Windbreaks and other woody buffers can provide protection from winter temperatures and winds if the hives are located on the leeward side, helping reduce winter mortality [94]. A study in

Kansas, USA, documented this protective service, with overwintered bee populations being up to 52% higher when hives were protected by windbreaks [95].

Foraging in moderate-to-high winds increases energetic costs for pollinators and can reduce pollination efficiency [88,96]. Agroforestry practices can reduce wind speeds, which increases pollinator efficiency and allows pollinators to forage during wind events that would normally reduce or prohibit foraging [42,91]. The protective effect on insect flight extends up to a distance equal to about nine times the height of the windbreak [97,98] and the sheltered zone contains higher numbers of pollinating insects [99], increasing pollination and fruit set [91,100].

### 3.4. Nesting and Egg-Laying Sites

The availability of nesting and egg-laying sites is an important element of pollinator habitat, although less is known about this component compared to foraging needs [20,101–103]. In general, pollinator populations benefit most from flower-rich foraging areas if suitable nesting or egg-laying sites are nearby [9].

Approximately 30% of native bees in North America are above-ground cavity-nesting species that build their nests in hollow tunnels in the soft pithy centers of twigs of some plants, in abandoned wood-boring beetle tunnels, or in tunnels where some species excavate themselves into wood [77,101]. Hedgerows have been shown to increase the availability of nesting sites for above-ground nesting species [22,104] and older hedgerows had a higher incidence of above-ground nesting bees [104]. A modeling study calculated a higher nesting potential for cavity-nesting species in landscapes with agroforestry compared to landscapes without agroforestry [105]. Incorporating woody species with soft pithy centers such as sumac (*Rhus* spp.), elderberry, and brambles may be beneficial in providing nesting sites [77,101].

The majority of North American native bees are solitary ground nesters that excavate underground tunnels for nesting, which can be negatively impacted by tillage in agricultural fields [106,107]. The presence of trees and shrubs may provide protected ground-nesting areas that have limited soil disturbance. Hedgerows may provide suitable ground-nesting habitat and increase diversity of ground-nesting bees [22,71,104]; however, another study did not find enhanced nesting rates for ground-nesting bees in hedgerows [102].

Bumble bees construct nests in small cavities, often in old rodent burrows, either underground or beneath fallen plant matter. They may select nest sites at the interface between fields and linear woody habitat such as hedgerows and windbreaks [108,109]. One study documented bumble bee nest densities twice as great in these linear woody habitats when compared with grassland and other woodland habitats [110], while another study found hedgerows to be less preferred when compared to herbaceous field margins and grasslands for nest-searching bumble bee queens [111].

Hedgerows and other agroforestry practices can provide egg-laying sites, larval host plants, and overwintering sites for lepidopteran (butterfly and moth) species [44,52,112]. In the U.S. mid-Atlantic region, woody species used as larval host plants were found to support 10 times more lepidopteran species than herbaceous plants [12]. Some of the most highly used plant genera by lepidopteran species include willows, birch, poplar, cherry, plum, and oaks (*Quercus* spp.) [12]. Lepidopteran species and other pollinators, including beetles, overwinter under bark and leaf litter found in hedgerows [44,112,113].

**Table 1.** The role of temperate agroforestry practices and woody plants on insect pollinators and pollination.

| Habitat Component or Ecosystem Service | Summary | Key References |
|---|---|---|
| Resins and oils | Honey bees harvest resins from tree buds, particularly poplar (*Populus* spp.), to make propolis, which provides antimicrobial and structural benefits for the colony. Other tree species, including pine (*Pinus* spp.), birch (*Betula* spp.), elm (*Ulmus* spp.), alder (*Alnus* spp.), beech (*Fagus* spp.), and horsechestnut (*Aesculus* spp.) are sources of resin. Some solitary bees also collect plant resins to include in brood cell linings and others use oils in brood cell provisions. | [75,76,78,82–84,86] |
| Early-season pollen and nectar | Woody species in temperate regions can provide important early-season sources of pollen and nectar. | [57,59–61,65,69,70] |
| Pollen protein quality | Willows (*Salix* spp.), cherry, and plum (*Prunus* spp.), and other woody species, offer pollen with high nutritive value. | [60,61,63–65] |
| Nectar sugar density | Tree and shrub flowers can provide nectar with relatively high sugar content and high flower densities. Hedgerows can provide greater nectar per unit area compared to woodlands and pastures. | [35,56–62] |
| Butterfly and moth larval hosts | Woody plants are important host plants for the larvae of many lepidopteran species (moths and butterflies). Some of the most highly used plant genera by lepidopteran species include oaks (*Quercus* spp.), cherry and plum (*Prunus* spp.), willows (*Salix* spp.), birch (*Betula* spp.), and poplar (*Populus* spp.). | [12,44,52,112] |
| Ground-nesting | Agroforestry practices can offer stable sites for ground-nesting bees and wasps in frequently disturbed agricultural landscapes. | [22,32,71,104,108–110] |
| Cavity-nesting | Shrub species with pithy centers such as elderberry (*Sambucus* spp.), sumac (*Rhus* spp), and brambles (*Rubus* spp.) can provide hollow tunnels for above-ground cavity-nesting bees. Dead trees and branches left in an agroforestry practice can also provide nesting sites. | [22,104,105] |
| Overwintering | Lepidopteran (butterflies and moths), Coleopteran (beetles), and other pollinators overwinter under bark and leaf litter found in hedgerows and other woody plantings. | [44,112,113] |
| Microclimate modification: wind | Windbreaks and other agroforestry practices can reduce winds and desiccation of pollen and floral parts, thereby enhancing pollinator foraging. Windbreaks can protect insect flight up to a distance equal to about 9 times its height. Agroforestry practices can help reduce winter mortality in honey bee hives by providing protection from winter winds. | [94,95,98,99] |
| Microclimate modification: temperature | Trees and shrubs can shade honey bee hives and reduce summer temperatures. Daytime air temperatures are several degrees warmer within a certain distance downwind of windbreaks and these elevated temperatures can increase pollinator activity and pollination if air temperatures at pollination time are below optimum. | [89,91,93] |
| Connectivity | Hedgerows and other linear agroforestry practices can facilitate pollinator movement across agricultural and urban landscapes at multiple spatial scales. These practices can provide spatially-distributed habitat that is within the foraging range of many pollinators, including short-distance foragers. | [22,53,104,114–120] |
| Barrier | Hedgerows may act as a barrier to pollinator dispersal and pollen transfer, depending on the landscape context and pollinator species. The orientation of plant rows may influence whether a hedgerow functions as a barrier or corridor. | [116,121–124] |

**Table 1.** *Cont.*

| Habitat Component or Ecosystem Service | Summary | Key References |
|---|---|---|
| Pesticide spray drift mitigation | Agroforestry practices can reduce pesticide exposure to pollinators by reducing spray drift from coming onto or leaving a farm by capturing particles and reducing wind speed. Windbreaks can reduce drift by up to 80% to 90%. Agroforestry buffers that are 2.5–3 m tall, with 40-50% porosity and fine, evergreen foliage are generally the most effective for drift prevention. | [125–133] |
| Pesticide runoff mitigation | Agroforestry practices can reduce pesticide exposure to pollinators by helping to capture pesticide runoff, prevent or slow pesticide movement through soil, and help break down some pesticides. | [134–138] |
| Refuge from pesticides | Agroforestry practices may serve as a safe haven for pollinators from pesticides, if adequately protected from spray drift. No-spray buffer zones may be necessary to protect the agroforestry planting. | [126,130,139–142] |
| Pesticide accumulation | Plants used in agroforestry practices can become contaminated with pesticides through aerial deposition and uptake through root systems. Plants contaminated by neonicotinoids through non-target drift of treated seed-coating dust during crop planting can negatively impact pollinators. | [143–146] |
| Adaptation to climate change | Agroforestry practices may offer ecological niches that allow pollinators to find suitable sites for thermal regulation under increasing temperatures and may serve as corridors and stepping stones to facilitate pollinator range shifts due to climate change. Landscapes that have a higher proportion of semi-natural habitats, including hedgerows and other woody plantings, may decrease the detrimental effects of warmer temperatures on pollinators. | [92,147,148] |
| Crop pollination | Agroforestry practices can provide increased pollination services and crop yields, including higher crop quality, although few studies have been conducted to document this direct agronomic benefit. | [30,46,149–151] |

## 4. Habitat Connectivity

Habitat fragmentation due to agricultural intensification, urban development, and other human activities is negatively impacting pollinators and pollination services at multiple spatial scales [9]. For example, Garibaldi et al. [152] estimated that fruit set of pollinator-dependent crops decreased by 16% at 1 km distance from the nearest pollinator habitat.

### 4.1. Site Connectivity

Based on field-level studies and modeling efforts, windbreaks, hedgerows, and alley cropping systems can provide pollinator habitat close to crops and at a scale that can benefit foraging [22,29,30,53,103,104,117–120,149]. For instance, Morandin and Kremen [22] observed higher native bee and honey bee numbers in fields adjacent to hedgerows than in fields adjacent to control edges. They also found a decrease in bee numbers as the distance from a hedgerow increased (i.e., 0 to 200 m), suggesting that habitat may need to be closely spaced across a field to promote exportation of bees into a field to support pollination [22]. Windbreaks are typically planted at intervals of 10 to 15 times windbreak height (H) to reduce wind erosion and enhance crop yields through microclimate modification [90,91,153,154]. For example, windbreaks 18 m tall may be spaced 180 m to 270 m across a field (center of field 90 m to 135 m from a windbreak), placing potential habitat within the foraging range of many pollinators, including short-distance foragers [155,156]. None of the studies reviewed found evidence that hedgerows or windbreaks reduced pollinators in crops or crop pollination by concentrating pollinators in the agroforestry practice. Habitat connectivity benefits can be higher when this semi-natural habitat is added to more homogenous and intensely managed fields and landscapes [21,157,158].

### 4.2. Landscape Connectivity

At the landscape scale, habitat connectivity is important for sustaining pollinator abundance, diversity, reproduction, and dispersal [9]. Agroforestry practices can support connectivity by serving as habitat corridors or stepping stones that facilitate pollinator movement across fragmented landscapes. Evidence documenting this function includes hedgerow-promoted movement of butterflies [123], moths [114], flies [53] and bees [115,116]; and butterfly travel along windbreaks [121] and riparian buffers [48]. A network of hedgerows was found to support wild bee species' richness and functional diversity [71] and the establishment and maintenance of populations at the landscape scale [157]. Syrphid flies were more abundant in forest-connected hedgerows than in forest edges with isolated hedges being intermediate [53]. Connectivity at the landscape scale may also benefit crop production as forest-connected hedgerows were documented to produce more high quality strawberries (*Fragaria x ananassa* Duch.) than isolated hedgerows [151]. Another study found a strong positive relationship between the percentage of hedgerows at the landscape scale (1 km radius) and pollinator visits, resulting in an increase in crop pollination of almost 70% when hedgerow cover increased from 1% to 6% [46].

### 4.3. Barrier

Agroforestry plantings may act as barriers to some pollinators, inhibiting movement between habitats [121,122]. Hedgerows may channel pollinator movement, which could enhance connectivity but restrict movement across hedgerows, isolating some plant populations [116]. The orientation of plant rows may influence hedgerows' abilities to promote movement or act as barriers [122,123] and keeping hedgerow height lower than 2 m may minimize the barrier effect on butterflies [45]. Pollen flow can also potentially be reduced across hedgerows [116] and possibly other tree-row plantings. Krewenka et al. [159] found that bee foraging was not impacted by hedgerows; however, another study found that bombyliid flies had reduced pollen transfer [124].

## 5. Pesticide Exposure Mitigation

Pesticides, particularly insecticides, can have acute toxicity leading to pollinator mortality and sublethal effects on growth, health, and behavior [160]. Across agricultural to urban landscapes, pollinators may come into contact with pesticides through numerous exposure pathways, including direct and residue contact, pollen and nectar contaminated by systemic insecticides, contaminated nesting materials, and contaminated water [9]. Agroforestry practices can affect the various exposure pathways and thereby influence the exposure and risk of pesticides to pollinators. Based on existing research, spray drift reduction and runoff mitigation are the primary ways that agroforestry can impact pesticide exposure.

### 5.1. Spray Drift Mitigation

Pesticide spray drift can be divided into thermal drift (lighter droplets transported to high altitude), vapor drift (volatilization from target), and droplet drift (droplets moved off-target by ambient wind) [129]. By reducing wind speeds and trapping particles, windbreaks and other linear agroforestry plantings can decrease pesticide droplet drift by up to 80%–90% and thereby reduce direct exposure to pollinators [126–129,137,142,161,162]. These practices can be used to minimize drift from either leaving or coming onto a site. Although agroforestry practices may influence thermal or vapor drift, no studies were found on this topic. Hedgerows and windbreaks that are 2.5 m–3 m tall, with 40%–50% porosity and fine, evergreen foliage have been shown to be the most effective for droplet drift reduction [125,127,131–133]. Hedgerows with porosity of nearly 75% have also been found to be effective in reducing drift by more than 80% [129]. Fine, evergreen, coniferous foliage can capture two to four times that of broadleaf species, with the additional benefit of trapping pesticides in early spring before deciduous plants have leafed out [126,131,132,142]. Leaves with hairy, resinous, and coarse surfaces can capture more particles than plants with smooth leaves [125,132].

### 5.2. Runoff Mitigation

In addition to spray drift, pesticides in solution, seed coating dust, or attached to soil particles can be transported off target by runoff [137]. Agroforestry practices can help reduce pollinator contact with pesticides by reducing runoff and contaminated surface water and by breaking down pesticides into less toxic forms. The perennial vegetation in agroforestry practices can help intercept pesticide-laden runoff, increase infiltration, and aid in phytoremediation of pesticides [24,134–138]. Based on a review of the available studies, Pavlidis and Tsihrintzis [134] documented a 40% to 100% reduction of pesticides (including herbicides) in runoff using agroforestry systems. A meta-analysis by Zhang et al. [136] highlights how sediment captured by vegetative buffers helps improve pesticide removal, particularly those pesticides that are strongly hydrophobic such as pyrethroids and many organophosphates. Plants and their rhizosphere microorganisms vary in their ability to degrade or immobilize pesticides. North American native trees with documented effectiveness in capturing pesticide run-off or immobilizing pesticides within their woody tissue include poplar, willow, birch, alder, black locust, and sycamore (*Platanus* spp.) [134,135].

### 5.3. Pesticide Accumulation

The same factors that make agroforestry practices effective buffers may lead to pesticide accumulation and pose danger for pollinators, particularly from systemic insecticides and those with long residual activity such as neonicotinoids [143,146]. Nectar and pollen of early-flowering woody species may become contaminated by systemic action of neonicotinoids or through non-target drift of treated seed-coating dust during crop planting [144]. Pesticide droplets, particles, or pesticides adhering to dust can also accumulate in the foliage or at the base of agroforestry practices [145], which pollinators may ingest or carry back to the nest [146]. This evidence suggests that if an agroforestry practice is to function as a buffer from pesticides with long residual activity, it will be

important to choose plants that are not attractive to pollinators. If long residual activity is not a concern, avoid using species that flower when pesticides are typically applied.

*5.4. Refuge from Pesticides*

Agroforestry practices may serve as refugia for pollinators and other beneficial insects if they are well protected from pesticides. No-spray buffer zones adjacent to agroforestry practice have been shown to be an effective strategy to protect these plantings from pesticide deposition [126,130,139–142]. In one study, spray drift deposition in hedgerows was reduced by 72% when a 12 m no-spray buffer zone was used next to the hedgerows [140]. At the landscape level, increasing the proportion of non-cropped habitat (i.e., riparian forest buffers, woodlands) in an agricultural area has been shown to buffer the effects of pesticides on native bees [163].

## 6. Crop Pollination Services

As described in the previous sections, scientific evidence demonstrates the conservation effects that linear agroforestry practices can provide to insect pollinators, including greater pollinator abundance and richness. Although these benefits should translate into enhanced pollination services leading to increased crop yields and quality, few studies have been conducted to document this direct agronomic benefit. Two studies demonstrated positive effects on canola yields due to hedgerows at local and landscapes scales [46,149], although these studies were conducted with potted plants. The observed yield effects might have resulted from concentration of pollinators and may not scale up to whole fields. For instance, one study found positive effects of hedgerows on pollination that did not result in increased yields in winter oilseed rape (*Brassica napus* L.) [30], while the other study showed no local effects on crop pollination in sunflower (*Helianthus annuus* L.) [34]. In apple orchards, researchers found increased pollinator abundance adjacent to an artificial windbreak, which led to a 20%–30% increase in fruit set with no reduction in fruit size [150]. While the artificial windbreak was created out of coir netting, this study may suggest potential yield increases due to pollinator activity in apple orchards with planted windbreaks.

Many factors are likely to influence the ability of agroforestry practices to promote crop pollination services, including specific pollinator attributes, field size, crop type, vegetative composition of the agroforestry practice, and landscape context [9]. The diversity of interacting variables makes it challenging to conduct studies and develop robust guidance for producers. For instance, the ratio of agroforestry practice to crop area in order to supply sufficient pollination service is largely unexplored [164]. One study demonstrated that native bees can provide full pollination services for watermelon (*Citrullus lanatus* Thunb.) when around 30% of the land within 1.2 km of a field is in natural habitat [165], which could be an approximate analog to an agroforestry practice. Regarding landscape context, one study found an increase in quality and quantity of strawberries grown adjacent to forest-connected hedgerows, as compared to isolated hedgerows or grass margins [151]. Plants placed at forest-connected hedgerows produced more high-quality strawberries with 90% classified as "marketable". In comparison, only 75% of strawberries from plants at isolated hedgerows, 48% of strawberries from plants on grassy margins, and 41% of strawberries from self-pollinated control plants were classified as marketable. Based on market prices of 2016, the increase in economic value between strawberries produced at grassy margins and forest-connected hedgerows amounted to 61% [151]. Cost-benefit studies that assess the benefits of an agroforestry practice for pollination services compared to the costs of installation and maintenance, opportunity costs, and costs of potential unintended negative effects are very limited. One study on hedgerows determined that seven years would be required for farmers to recover implementation costs based on the estimated yield benefits from both pollination and pest control to the crop [149]. With numerous interacting ecosystem services, cost-benefit analyses should consider the suite of agronomic services in order to provide comprehensive economic assessment.

## 7. Conclusions

Agroforestry has been identified as a multifunctional land use approach that can balance production with environmental stewardship [166]. Combining woody vegetation with cropping and livestock production via agroforestry systems increases total production, enhances food and nutrition security, and protects natural resources, while helping to mitigate and adapt to the effects of climate change [23,166]. Capitalizing on insect-based ecosystem services, agroforestry offers emerging opportunities to support pollinators and other beneficial insects and their services including crop pollination and biological pest management. Based on the available scientific literature, linear agroforestry practices (i.e., windbreaks, hedgerows, riparian buffers, alley cropping) in temperate regions can aid pollinator conservation by providing habitat, including foraging resources and nesting or egg-laying sites, enhancing site and landscape connectivity, and mitigating pesticide exposure. This evidence provides considerations for enhancing agroforestry in supporting pollinators such as the importance of early- to mid-season flowering woody species to provide pollen and nectar. From this review, efforts are currently underway by the Xerces Society for Invertebrate Conservation and the USDA National Agroforestry Center to develop regionally-based recommendations for agricultural producers.

There are still important knowledge gaps on how to design and manage these agroforestry practices to deliver pollinator conservation services. Key gaps to investigate include optimized design and placement of agroforestry practices to benefit pollinators (e.g., best plant combinations to provide foraging resources in space and time, orientation, and distance to target crop), designs to support habitat connectivity (e.g., spacing within the farm landscape, minimizing potential barrier effects), and ongoing maintenance for the long-term health of the practice, balanced with the impacts of management on pollinators (e.g., efforts to increase nesting habitat, effects of brush removal). Regarding silvopasture and forest farming practices, the knowledge base is very limited in temperate regions and could benefit from initial research that assesses opportunities and constraints of providing pollinator conservation benefits with these forest-based practices. For silvopasture, the interactive effects of stocking rate/timing, canopy density, and plant selection will be important variables to assess; while in forest farming, a better understanding of pollination needs of many forest-farmed crops is particularly needed. Although limited, some evidence suggests that agroforestry practices could accumulate pesticides and may inadvertently expose pollinators to higher concentrations. Future investigations should address this concern by better documentation of exposure risk and translocation of different pesticides in plant tissues, nectar, and pollen of agroforestry species to determine if some woody plants pose a greater risk. In addition, research on agroforestry species that best support phytoremediation processes that break down pesticides and analyses of agroecology approaches that combine best management practices reducing pesticide use and exposure along with other farm goals is needed.

Based on this review, agroforestry practices with appropriate species and management can support better pollinator conservation than homogenous fields and farms without agroforestry. However, for producers in temperate regions interested in relying on agroforestry to provide full crop pollination services from insects, agronomic evidence is currently limited for decision making. Crop type, landscape context, plant material used in the agroforestry practice, ratio of agroforestry area to crop area, and spatial distribution of the agroforestry practice are critical variables needing further research in order to better understand the financial impacts of agroforestry on pollination services. Although investigating pollination services is challenging, this information will be key for advancing the use of agroforestry to support this ecosystem service. In addition, cost-benefit studies that assess the economic benefits of an agroforestry practice for crop pollination compared to implementation, management, and opportunity costs need to be conducted. A primary advantage for using agroforestry to support pollinators is that these practices are often being implemented for other production and ecosystem services. These practices often inherently provide some pollinator benefits and with

additional considerations during design and management, the effectiveness of agroforestry practices for pollinator conservation and pollination services should be enhanced.

**Author Contributions:** G.B. conceived and designed the review. G.B., J.H. and N.L.A. collected the literature and conducted the systematic review of available data on the subject matter. G.B., J.H. and N.L.A. wrote the first draft of the manuscript. M.V. reviewed and helped revised various versions of the manuscript.

**Funding:** This work was supported by funding provided by a contribution agreement with the United Stated Department of Agriculture (USDA)-Natural Resources Conservation Service (NRCS) to the Xerces Society.

**Acknowledgments:** We would like to acknowledge David Inouye and two anonymous reviewers whose comments and feedback helped to improve the paper.

**Conflicts of Interest:** The authors declare no conflict of interest.

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
