# Peer review of "Temperate Agroforestry Systems and Insect Pollinators: A Review"

_forests, doi:10.3390/f10110981_

Round 1
Reviewer 1 Report
The manuscript is a nice contribution to the pollination literature. It's well written, and includes the relevant literature (although I've suggested some references about coffee pollination that I think should be added). I've made some editorial suggestions/corrections on the PDF.

Author Response
Thank you for your thoughtful review and we appreciate the suggested references regarding coffee pollination. There is a rich literature base regarding this topic. Since our focus is on temperate agroforestry systems, we decided not to include studies on coffee. Hopefully a review on tropical agroforestry systems and pollinators will be conducted in the near future.
Thank you for the editorial comments provided on the marked-up PDF. We have incorporated these changes.
Reviewer 2 Report
An excellent, well written, and comprehensive review of the topic of temperate agroforestry and insect pollinators. The authors provide a timely review of current information and identify gaps in current knowledge that need additional research.
I have not personally used the PRISMA review methodology, so I can't comment specifically on their use of it (e.g., methods used to screen papers, etc.). However, based on my knowledge of the topic, the authors successfully identified the relevant literature to include in the review. I did not check the reference list for formatting.
The paper well written and easy to follow. Although I didn't specifically edit the manuscript, I did not notice any typographical mistakes with the exception of line 5, the author affiliations, which should read, "...Lincoln, NE..." instead of "...Lincoln and NE..."
Author Response
We appreciate your comments and have corrected the mistake on line 5.
Reviewer 3 Report
The authors prepared a concise review of the literature related to pollinators and agroforestry. I think with minor changes this will provide a valuable contribution documenting the current state of knowledge. The key issue is staying focused on the topic at hand and being more specific in conclusions and providing more synthesis rather than summary.
Abstract
The abstract does a nice job summarizing the manuscript findings and staying focused.
Introduction
The introduction is to general. The first 3 paragraphs have nothing to do with the topic at hand and could be pasted onto any manuscript about pollinator conservation (but shouldn’t be). The authors do not need to espouse the many benefits of pollinators. A sentence or two would set the stage. Is it really pertinent to agroforestry that bees are required to produce crops that provide essential nutrients? Likewise, with the risk paragraph (ln51-56). Keep it focused on agroforestry. Bees are good, they are in trouble, we got it.
The introduction would benefit from more information about agroforestry so we understand what it is and how it is used. Benefits to pollinators are a secondary benefit what are the primary benefits? What tree species are common? The intro should provide context for understanding the rest of the paper.
Ln151-159 This section does not mention agroforestry. It teaches us that bees use and collect resins and some of the reasons such as reducing disease transmission. We also learn some tree species that provide resins. Please bring this back to agroforestry. Are these tree species commonly used in agroforestry? Should they be? Do they have the traits necessary for primary benefits sought from agroforestry?
Ln300-311 This section contains information on agroforestry but none on bees. How does pesticide runoff, or preventing runoff, affect bees? I’m not sure but this section should tell me.
Conclusions
This section could be more specific with the type of research that is needed. There is mention of knowledge gaps followed by a list of topics which are vague. It would be more valuable to have some specific recommendations or priorities for what next steps are needed. Likewise ln384-388 tells use some risks may be posed by agroforestry but more research and acomprehensive approach could help minimize them. Be more specific. Synthisize the information you reviewed and make some recommendations. If a new grad student was reading this, looking for inspiration, what would they take from it. You could have an influence on what work comes next.
The final paragraph is also very vague. I list of more work to be done but no synthesis. There must be some recommendations for producers. Is agroforestry generally better or worse then fields with no trees? Are there tree species that are better than others? If someone called Xerces Society asking for help designing their field would you say “For producers interested in using agroforestry in temperate regions to support insect-based crop pollination, agronomic evidence is currently limited for decision making. Crop type, landscape context, plant material used in the agroforestry practice, ratio of agroforestry area to crop area, and spatial distribution of the agroforestry practice are some of the variables needing further research.” Of course not you would say based on available evidence the best approach seems to be…..and make some recommendations.
Author Response
Introduction
Thank you for your very constructive review. We reduced the first 3 paragraphs to one paragraph based on the comments. Since Forests is a cross-disciplinary journal and that the readership is not likely to have a pollination or entomology background, we decided to retain some background information to set the stage. This is keeping in style with other review manuscripts published in Forests.
We also incorporated more information on agroforestry to create better context for understanding the rest of the paper (ln49-60).
Ln151-159 We added new material to tie these species back to agroforestry.
Ln300-311 We add additional sentences to the runoff mitigation section to make a better connection of how this function reduces exposure risk for pollinators.
Conclusions. We added more detailed recommendations for next steps in research including new research recommendations regarding pesticide issues and silvopasture/forest farming. We appreciate your comments about having more recommendations to provide for producers. Considerations are found throughout the paper (e.g., willows (Salix spp.) offer pollen with high nutritive value, design characteristics of spray drift buffers). As a next step, we are developing regionally-based recommendations that will be targeted for the producer audience and that will include more specific information (e.g., listing willow species that are appropriate for a particular region). These guidelines will published in the future.